# Characteristics of the Family Support Network of Pregnant Adolescents and Its Association with Gestational Weight Gain and Birth Weight of Newborns

**DOI:** 10.3390/ijerph16071222

**Published:** 2019-04-05

**Authors:** Reyna Sámano, Hugo Martínez-Rojano, Gabriela Chico-Barba, Bernarda Sánchez-Jiménez, Daniel Illescas-Zarate, Ana Lilia Rodríguez-Ventura

**Affiliations:** 1Departamento de Nutrición y Bioprogramación, Instituto Nacional de Perinatología, Secretaría de Salud Montes Urales 800, Miguel Hidalgo, Lomas Virreyes, Ciudad de Mexico CP. 11000, Mexico; ssmr0119@yahoo.com.mx (R.S.); gabyc3@gmail.com (G.C.-B.); rovalilia@hotmail.com (A.L.R.-V.); 2Departamento de Posgrado e Investigación, Escuela Superior de Medicina del Instituto Politécnico Nacional, Plan de San Luis y Díaz Mirón s/n, Colonia Casco de Santo Tomas, Delegación Miguel Hidalgo, Ciudad de Mexico CP. 11340, Mexico; 3Escuela de Enfermería, Universidad Panamericana, Augusto Rodin 498, Insurgentes Mixcoac, Alcaldía Benito Juárez CP. 03920, Mexico; 4Subdirección de Investigación en Intervenciones Comunitarias, Instituto Nacional de Perinatología, Montes Urales 800, Lomas Virreyes, Alcaldía Miguel Hidalgo CP. 11000, Mexico; emiberna20@yahoo.com.mx; 5Centro de Investigación en Nutrición y Salud (CINyS) del Instituto Nacional de Salud Pública. Avenida Universidad 655, Santa María Ahuacatitlán, Cuernavaca, Morelos 62100, Mexico; daniel_i_z@hotmail.com

**Keywords:** adolescent pregnancy, family support network, birth weight, gestational weight gain

## Abstract

It has been proposed that, in the Mexican culture, family support can be a factor that contributes to protect the maternal and child health of pregnant adolescents. There may be complex associations between family support and the circumstances of a pregnancy during adolescence. The aim of the study was to analyze the association between the family support network (FSN) characteristic and the maternal and neonatal outcomes in Mexican adolescents. A cross-sectional study was conducted, and 352 pregnant adolescents participated; their FSN during pregnancy was assessed. The gestational weight gain and birth weight/length of newborns were registered. The size of the FSN was described and divided into quartiles; the main members for each quartile were identified. Then, sociodemographic and clinical variables were compared by FSN quartiles. Logistic regression models were performed to assess the association of FSN size and pregnancy and neonatal outcomes. Our results indicate that the mean age was 15 ± 1 year old. The primary support member in the FSN was the mother of the adolescent in each quartile, except for quartile 3, where the primary support was the mother-in-law. In quartile 3 there was a significantly lower gestational weight gain compared to quartile 4 (11.8 ± 5 vs. 13 ± 5 kg, *p* = 0.054). According to the regression model, a higher risk of small for gestational age (OR 2.99, CI 95% 1.25–7.15) newborns was found in quartile 3. We conclude that the maternal and neonatal outcomes did not differ between quartiles of FSN size, except for quartile 3. Small for gestational age newborns were observed when a non-blood relative was present in the FSN. The quality rather than the network size might be more important for improving pregnancy outcomes.

## 1. Introduction

Social capital is composed by several elements of social organization, such as norms and networks [1]. The social support network includes a combination of actions that are essential for maintaining individual physical and psychological health [2,3]. These actions are categorized into emotional (offering company and talking about problems), material (loaning money or things), collaborative (doing housework or providing transportation), and/or informational (providing the address or phone number of a medical service) support. Social support can be offered by family, friends, neighbors, or others. Additionally, it can be assessed by its size, function, and quality of support provided [4].

During pregnancy, there is evidence that family support has beneficial effects on pregnancy and birth outcomes, like postpartum depression [5], adequate infant birth size [6], breastfeeding [7], and infant adiposity [8]. When pregnancy occurs during adolescence, family support becomes more important, because adolescents have a higher risk of inadequate gestational weight gain and low birth weight, in addition to their owns needs as adolescents per se [9,10].

It is known that family support has a positive impact on pregnancy in adolescence, especially when this support is provided by a female relative, such as the mom or older sister [11,12], but there are other family members that offer support to the adolescent [13]. Furthermore, the evidence focuses on the function and quality of support provided by the family network members and does not focus on the family support network size. Therefore, the aim of the present study was to analyse the association between the family support network (FSN) characteristics (size and members) and the outcomes of pregnancy in adolescents.

## 2. Materials and Methods

### 2.1. Study Design and Setting

This was a cross-sectional study conducted from 2008 to 2014 that included adolescents from 12 to 18 years old. The sample was representative of those adolescents who sought and received prenatal medical care in the National Institute of Perinatology (Instituto Nacional de Perinatología, INPer) in Mexico City.

### 2.2. Participants

The age range of the participants was between 12 and 18 years. At the lower limit age, 12 years was the youngest age of pregnant adolescents that sought medical care at the INPer and met the inclusion criteria of the present study. At the upper limit age, we set 18 years of age because the social factors for 19-year-olds are different; this is the first year in which most adolescents are no longer in high school and either start a job or enter university. Even if the adolescents have already dropped out of school, most of their friends are of the same age.

The inclusion criteria were adolescent women, primiparous, with a singleton and healthy pregnancy and healthy delivery, who attended INPer for medical care. Excluded from the study were five adolescents giving birth in another hospital and six whose neonates were diagnosed with some disorder or complication related to intrauterine growth restriction or abnormal growth (myelomeningocele, gastroschisis, or oligohydramnios). There were also eight adolescents excluded because they developed pregnancy-related illnesses, including diabetes and preeclampsia. Other exclusion criteria were related to adolescents with a drug addiction, legal problems (an added stress factor) or a strict vegetarian diet (an additional factor that affects weight gain). There were no cases of the latter three criteria.

### 2.3. Measurement of the Family Support Network (FSN)

We analyzed the forms of social support, which were defined together as an informal network constituted by a group of individuals who communicate with each other due to personal ties outside any institutional or organizational context. Hence, the social network is not always strictly comprised of family. This social network constitutes the most robust support that keeps an individual linked to society in an autonomous, integral, and independent form.

### 2.4. Evaluation of the FSN Size

The FSN size was explored during the third trimester of gestation, using the Nava–Aguilar and Terán–Trillo instrument [14,15]. The questionnaire allowed the collection of data and examination of the different egocentric networks of formal and informal social support. For the current investigation, only the information corresponding to informal support was taken into account.

The instrument utilized to assess the family support networks is the only one validated for an adolescent population in Mexico. Although the instrument is designed to identify the main members who offer support to adolescents in different situations, it does not evaluate family functionality. The first section was used for the collection of personal data of the ego (pregnant adolescent), including name, age, marital status, level of education, occupation, number of people in the home, and family type. The second section collected data on people who provided social support, such as their age, name, and relationship to the ego. Four types of support were listed in four columns: Emotional (affection and trust), informational (about medical services), collaborative (providing help and keeping one company), and material (money or things). The ego mentioned the type of support provided by each member of her FSN and the researcher wrote it down in the appropriate column. As a result, the network size and member characteristics were identified based on information about the relationship, gender, and age. In addition, we obtained information about the number of children and adolescents (≤15 years old) who lived in the same house as an adolescent mother.

The FSN size was obtained by counting the members who offered any type of help to the ego. Then, it was divided into quartiles as follows: Q1 (*n* = 136) with 1 member, Q2 (*n* = 84) with 2–4 members, Q3 (*n* = 59) composed by 5 members; and Q4 (*n* = 73) with 6–10 members.

### 2.5. Anthropometric Evaluation

Anthropometric measurements were obtained during the third trimester of pregnancy. The pre-gestational weight (kg) was determined by asking each participant about her weight three months before pregnancy; there is a high correlation between the self-reported and real weight [16]. Height (cm) was measured with a stadiometer (SECA, model 208) having an accuracy of 0.1 cm. Based on weight and height, the pre-gestational body mass index (BMI) was calculated by dividing pregnancy weight in kilograms by height in squared meters (weight/height^2^), and then categorized as follows: Underweight (≤18.5), normal (18.5–24.99), overweight (25–29.99), and obese (≥30) [17].

The gestational weight gain was obtained by subtracting the pre-gestational weight from the final gestational weight that was measured one week before childbirth using a digital scale (TANITA model BWB-800), with an accuracy of 100 g. The gestational weight gain was considered normal according to the following criteria: 12.5 to 18 kg in underweight women (BMI ≤ 18.5), 11.5 to 16 kg for women with normal weight (BMI 18.5–25), 7 to 11.5 kg for overweight (BMI 25–29.99), and 5 to 9 kg in obesity (BMI ≥ 30) [18]. Trained and standardized personnel performed the measurements using the *Lohman* technique [19].

### 2.6. Neonatal Characteristics

Gestational age was calculated according to the date of the last menstrual period and then was corroborated by ultrasonography. Gender, weight, and length were obtained from the birth certificate and medical records. Then, the percentile of weight/length for gestational age was calculated, and in accordance with Fenton et al. [20], the percentile was categorized as follows: Small for gestational age (under the 10th percentile), adequate for gestational age (between the 10th and 90th percentile), or large for gestational age (above than the 90th percentile).

### 2.7. Sociodemographic Data

Age was noted in years. Level of education was categorized into those having <9 years of education; this number refers to junior high school in the Mexican education system. Occupation was classified as student, worker, and others. Marital status was classified as single and married/cohabiting. Type of family was classified as nuclear (when the adolescent mother lived with her parents and siblings); extensive/blended (when the adolescent mother lived with her parents and grandparents; and single parent (when the adolescent mother lived only with either her mother or father). Furthermore, the number of prenatal medical consultations and prenatal care initiation were obtained by medical records. Gynaecological age was calculated by the difference between chronological age and menarche age; both in years. Socioeconomic level was assessed by using a validated questionnaire, which consisted of 10 questions about the goods that the family owns, its recreational activities, services contracted, and other factors, assigning the adolescent to one of six of socioeconomic levels: Extremely poor, very low, low, medium, medium-high or high [21].

### 2.8. Dietary Evaluation

The habitual energy intake was assessed in the third trimester of pregnancy, using a food frequency questionnaire of 110 items. This questionnaire was previously validated by Hernández-Ávila et al. in a Mexican population [22].

### 2.9. Ethical Considerations

The Institutional Review Board of the National Perinatology Institute approved the study. The purpose of the study was explained to the participants, and all questions were clarified. All participants, as well as their parents or legal tutors, signed an informed consent. This study complies with the Helsinki Declaration about research on human subjects [23].

### 2.10. Statistical Analysis

We performed a descriptive analysis of the characteristics of the study population. Frequencies and percentages were calculated for categorical variables and mean and standard deviation for continuous variables.

Regarding the bivariate analyses, the size of the FSN was divided into quartiles. The chi-squared test was used to compare quartiles of FSN and categorical variables, and the ANOVA test was used when analyzing quantitative variables (normal distribution with Bartlett’s test, *p* > 0.05). Adjustments were made using the *Bonferroni* for multiple comparisons. Variables not showing a normal distribution were analyzed with the Wilcoxon-type test.

A logistic regression model was performed to obtain the odds ratio with 95% confidence intervals for the associations between the size of FSN and the outcome small for gestational age. Dummy variables were created for the number of members of the support network during pregnancy as the primary independent variables. The model was adjusted for potential confounding factors including pre-gestational weight (kg), the age of the adolescent (years), level of education, energy intake (kcal), and gestational weight gain (low, normal, or excessive). Statistical significance was considered at 5% level. All statistical analyses was carried out using STATA for Windows, version 12. StataCorp. 2011. Stata Statistical Software: Release 12. College Station, TX: StataCorp LP. TX, USA.

### 2.11. Ethics Approval and Consent to Participate

Each adolescent and their parents or legal tutor signed an informed consent and acceptance letter and accepted the sharing and publishing of the results without specifying participants’ names. The study has been approved by the Research and Ethical Committee of the National Institute of Perinatology (Number of registration 212250-49481).

## 3. Results

### 3.1. Sociodemographic Data

A total of 352 pregnant adolescents participated in the study. Mean age was 15 ± 1 year old, and mean gynaecological age was 4 years. Regarding the home environment, 57% of the participants came from nuclear families, 28% from extended or blended families, and 15% from single-parent families. All adolescents were from a very low, low, and medium socioeconomic level. Over 77% of the participants had less than 9 years of schooling; this means that they did not finish middle school. Adolescents of quartile 3 had a higher frequency of being married or cohabiting. The mean number of prenatal medical consultations was 7 ± 2, with prenatal care starting, on average, at week 19 of gestation (the second trimester). A total of 48% of the pregnancies ended by caesarean section and 52% by vaginal birth; these numbers were similar in all quartiles (Table 1). We observed that it was common to find children younger than 15 years in the family, especially in Q1 compared to the other quartiles, without statistical significance; Q1: *n* = 55 (40.5%), Q2: *n* = 25 (29.6%), Q3: *n* = 18 (29.8%), Q4: *n* = 17 (23.7%), *p* = 0.106.

### 3.2. Family Support Network

In all the quartiles, the person most frequently identified by the adolescent as her main support during pregnancy was her mother, followed by her partner and her father. The exception was in quartile 3, in which the mother-in-law in some cases provided the main support, and the father had no representation (*p*-value: 0.001, Chi-squared test (Figure 1)). The types of support mentioned by most of the participants were affection (329 (93%)), trust (312 (90%)), and information (239 (69%)). Material support showed the lowest frequency of 190 (55%) in all quartiles. No significant difference was found between quartiles on these percentages.

### 3.3. Dietary Evaluation

Energy intake by the adolescents was over 2000 kcal in all quartiles, with a tendency to a higher intake by the adolescents in the fourth quartile (compared to all other quartiles; *p* = 0.020). The difference between the fourth and third quartile was 176 kcal (Table 2).

### 3.4. Evaluation of the Newborn

The weight and length of the newborns were less for the participants in the third quartile, although the difference with the other quartiles was not significant. Gestational weight gain was lowest in the third quartile compared to the other groups, with a difference of at least 1 kg, (*p* = 0.003) (Figure 2). In this same quartile, there was a higher proportion of newborns that were small for the gestational age (*p* = 0.010). The frequency of newborns with an adequate birth weight for the gestational age was the highest in quartiles 2 and 4 (Figure 3 and Table 2).

According to the logistic regression model, the adolescents of the third quartile had a three-fold higher risk (OR, 3.03; CI 95%, 1.3–7.0) of having a newborn that was small for gestational age. When adjusting the model for maternal age, pre-gestational weight, level of education, energy intake, and gestational weight gain, the result was similar and significant when comparing quartile 3 to the other quartiles (OR, 2.99; CI 95% 1.25–7.15) (Table 3).

## 4. Discussion

### 4.1. Family Support Network Members

The main member identified in the adolescent’s family support network was the mother, followed by the partner. The type of support that was most frequently provided was affection and trust, and when the family support network lacked blood relatives, such as the adolescent’s father, there was a higher risk of having a small for gestational age baby.

On the other hand, the role of mothers of pregnant adolescents is known to have an impact on their nutritional status, in the encouragement of using contraceptives, in the prevention of a second pregnancy, and in the emotional support during [24] and after pregnancy [25]. When the adolescent girl was living apart from her parents, the support of the mother could have been limited by the intervention of the mother-in-law and/or husband.

### 4.2. Family Support Network Size

Other authors have demonstrated that the size of a support network is not always related to a positive impact on pregnancy and birth outcomes. For instance, Jones et al. found that there was no relationship between family support and mental health in a group of pregnant women experiencing interpersonal violence [26].

The current results contrast with those reported in previous studies that used different methods, as well as indirect methods to evaluate the size of the support network [27].

### 4.3. Gestational Weigh Gain and Birth Weight

We showed that the FSN size was not associated with gestational weight gain or birth weight, which agrees with a previous study in pregnant adults. In contrast, Elsenbruch et al. linked social support to a prenatal and postnatal outcome, observing that the greater the social support, the greater the length of the newborn [28].

Quartile 3 had an intermediate number of members in their FSN, and there was a lower gestational weight gain and lower neonate birth weight. A possible explanation is the inadequacy of the family dynamics [27]. In this quartile, the participation of the mother in-law was more frequent instead of the parents. Fernandez et al. [29] found that the presence of strong family support in the Mexican culture, especially centered on the adolescent’s mother, is a crucial factor for the protection of maternal and child health. Similar results were reported in a sample of Mexican origin adolescent mothers in the United States [30].

Our results do not support, but neither do they refute that the FSN is more effective when the mother and father are the main sources of support. Kumar et al. observed that when the mother of the pregnant adolescent supported her daughter during pregnancy and postpartum, there were better maternal and neonatal outcomes [12].

The energy intake of the participants was not associated with the weight of the newborn but was indeed related to the size of the FSN, as was demonstrated when comparing the average amount of energy intake within all quartiles, where the energy intake was higher in quartile 4 (*p* = 0.020). This could be due to the positive effect exerted by a larger FSN on the quantity and quality of the diet [31], as well as on other factors related to the consumption and expenditure of energy, for instance stress [32], which could be common in low-income adolescent women.

In the current study, most of the participants were in a very low or low socioeconomic level, where social disadvantages are common, such as educational lag, no health insurance coverage or being an informal worker [33,34,35,36]. There is evidence that a low socioeconomic level negatively influences perinatal outcomes [37], which may be due to a poor family support network [38]. Jonas et al. demonstrated, in a group of adolescents from South Africa, that poverty and lack of parental support was associated with a high number of pregnancies and negative health results during pregnancy [39].

### 4.4. Implications of the Study

The present research highlights the importance of the size and type of members of the family support network. The main members of the FSN are the mother and father, and when any of them are absent it might negatively affect pregnancy, but this is not the only factor associated.

Further studies should include qualitative aspects of the FSN to create an appropriate intervention in pregnant adolescents.

### 4.5. Limitations and Strengths of the Study

The main limitation of the present study is regarding the cross-sectional design, which did not allow for the corroboration of the continuity of the FSN during the entire pregnancy. Another limitation is the instrument used to assess the family support network, which is designed to identify the main members supporting the adolescents in different situations but is deficient in elements that evaluate family functionality and quality.

The strength of the present study is the sample, which represents the low or low-medium socioeconomic level; it also has similar nutritional characteristics and was comprised mostly of adolescents with a healthy pregnancy.

## 5. Conclusions

We conclude that maternal and neonatal outcomes, especially gestational weight gain and birth weight, did not differ between different sizes of a family support network, except when a family network was comprised of five members. The presence of small for gestational age newborns was more frequent when the main support was provided by a non-blood relative. The quality rather than the network size might be more important for improving pregnancy outcomes.

## Figures and Tables

**Figure 1 ijerph-16-01222-f001:**
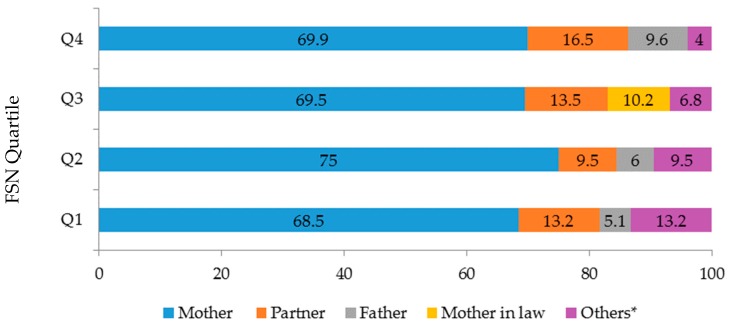
Members identified as the primary support during pregnancy, according to each quartile. Q: quartile; * includes siblings, grandparents, uncles and aunts, cousins and friends.

**Figure 2 ijerph-16-01222-f002:**
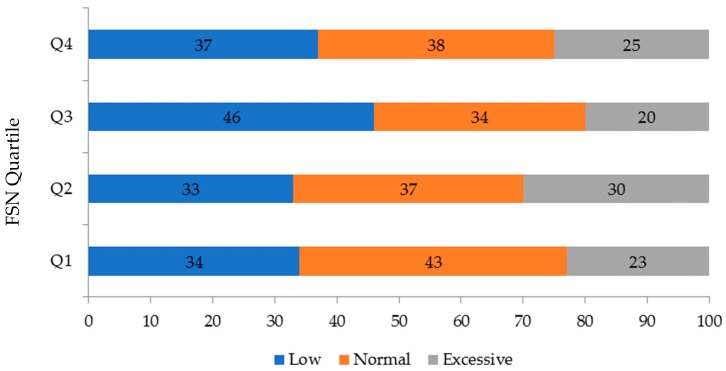
Gestational weight gain according to family support network quartile.

**Figure 3 ijerph-16-01222-f003:**
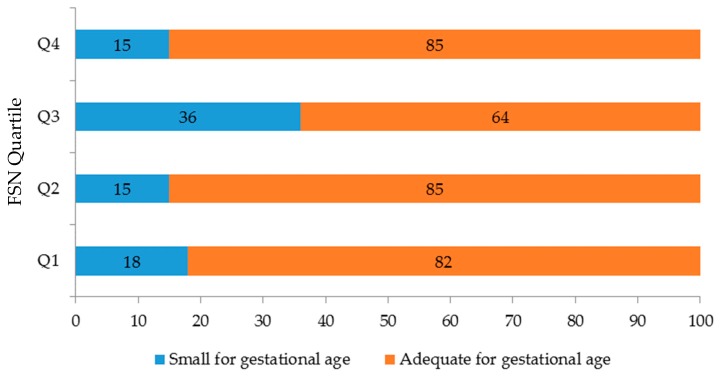
Birth weight according to family support network quartile.

**Table 1 ijerph-16-01222-t001:** General characteristics of the pregnant adolescents according to family support network quartile (*n* = 352).

Characteristics	Q1 (*n* = 136)	Q2 (*n* = 84)	Q3 (*n* = 59)	Q4 (*n* = 73)	*p*-Value
Age (years) *	15.3 ± 1	15.6 ± 1	15.7 ± 1	15.4 ± 1	0.360
Age of menarche (years) *	11.7 ± 1	11.7 ± 1	11.5 ± 1	11.9 ± 2	0.450
Gynaecological age (years) *	3.6 ± 1.4	3.9 ± 1.5	4.2 ± 1.5	3.5 ± 1	0.900
Level of education **					0.320
<9 years	117 (86)	65 (77)	44 (79)	60 (85)	
≥9 years	19 (14)	19 (23)	12 (21)	11 (15)	
Partner’s occupation **					0.650
Student	38 (29)	21 (27)	12 (23)	13 (20)	
Working	69 (52)	46 (60)	32 (62)	43 (66)	
Other	24 (18)	10 (13)	8 (15)	9 (14)	
Marital status (*n* = 341) **					0.016
Single	88 (66)	47 (57)	23 (42)	35 (51)	
Married/cohabiting	46 (34)	36 (43)	32 (58)	34 (49)	
Number of persons living in the household **					<0.001
0–2	59 (43)	17 (21)	13 (23)	13 (18)	
3–4	51 (38)	44 (52)	26 (46)	33 (46)	
5–24	26 (19)	23 (27)	17 (30)	25 (35)	
Prenatal care initiation (weeks) *	19 ± 6	19 ± 6	19 ± 5	18 ± 6	
Prenatal care initiation (by trimester) **					
First	23 (17)	16 (19)	11 (20)	15 (21)	0.790
Second	107 (79)	65 (77)	40 (71)	53 (75)	
Third	6 (4)	3 (4)	5 (9)	3 (4)	

^(^*****^)^ Mean and standard deviation. ^(^******^)^ Frequency (%).

**Table 2 ijerph-16-01222-t002:** Maternal and neonatal characteristics by quartiles of the family support network.

Characteristics	Q1 (*n* = 136)	Q2 (*n* = 84)	Q3 (*n* = 59)	Q4 (*n* = 73)	*p*-value
**Mother**					
Pre-gestational weight (kg)	51.2 ± 7.9	52.2 ± 8.8	52.5 ± 7.5	52.0 ± 7.6	0.670
Pre-gestational BMI (kg/m^2^)	21.2 ± 3.3	21.8 ± 3.9	21.7 ± 3.3	21.5 ± 3.3	0.640
Energy intake (Kcal)	2035 ± 572	2127 ± 573	2124 ± 513	2300 ± 592	0.020
Gestational age (weeks)	39 ± 1	39 ± 1	38 ± 2	39 ± 1	0.540
Total gestational weight gain (kg)	12.8 ± 4.9	13.1 ± 6.0	11.8 ± 4.6	13.0 ± 4.9	0.054
**Newborns**					
Gender *					
Girl	74 (54)	47 (56)	32 (57)	34 (48)	0.690
Boy	62 (46)	37 (44)	24 (43)	37 (52)	
Birth weight (g)	2897 ± 456	2964 ± 396	2790 ± 483	2914 ± 440	0.150
Birth weight for gestational age (*z*-score)	−0.62 ± 0.9	−0.46 ± 1.0	−0.83 ± 0.8	−0.59 ± 0.84	0.120
Birth length for gestational age (*z*-score) ^†^	−0.35 ± 0.8	−0.2 ± 1.0	−0.55 ± 0.9	−0.39 ± 0.88	0.140
Head circumference (cm)	33.2 ± 1.4	33.1 ± 1.7	33.0 ± 1.3	33.9 ± 2.0	0.700

Values are shown as the mean ± standard deviation. ^(^*****^)^ Frequency (%). Chi-squared test. ^†^ Five cases of large newborns, relative to gestational age, are included (percentile >90 for the gestational age and gender).

**Table 3 ijerph-16-01222-t003:** Associations between small for gestational age newborns and family support network (in quartiles) provided to pregnant adolescents.

	Odds ratio	*p*-value	95% CI
Model 1 *			
Quartile 1	1.23	0.60	0.57–2.67
Quartile 2	0.99	0.99	0.42–2.39
Quartile 3	3.03	0.01	1.30–7.0
Quartile 4	Reference		
Model 2 **			
Quartile 1	1.1	0.81	0.49–2.44
Quartile 2	0.89	0.80	0.36–2.20
Quartile 3	2.99	0.01	1.25–7.15
Quartile 4	Reference		

^(^*****^)^ Crude model. ^(^******^)^ Model adjusted for age (years), pre-gestational weight (kg), level of education (years), energy intake (kcal), and gestational weight gain (low, excessive). CI: confidence interval.

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
