# Peer review of "Characteristics of the Family Support Network of Pregnant Adolescents and Its Association with Gestational Weight Gain and Birth Weight of Newborns"

_ijerph, 2019, doi:10.3390/ijerph16071222_

Reviewer 1 Report

The main objective of this manuscript was to investigate the associations between the size of family support network of pregnant adolescents and its association with gestational weight gain and birth weight of newborns. This manuscript will contribute to the existing literature of the size of family support network of pregnant adolescents and adverse birth outcome. However, the manuscript is difficult to follow as written, and the results are not clear as presented. Also, there are many grammatical errors. Please have the manuscript edited by a native English speaker who can also help you improve the scientific presentation as well as the grammar. In addition, there are also many errors about format in the paper, please check your manuscript carefully.

 Major comments

1.Abstract:

1)    method: The describe of population is not clear. Further details about the total number of study population have recruited in the study should be given on the method.

2)    method: Please correct “The size the FSN was identified for each 23 participant and divided into quartiles…”

3)    results: “The results indicate that the mean age was 15 ±1.26 year…” the author should add the detailed information to make sure the reader could capture the exact information.

4)     conclusion: “The In this quartile, there was a significantly lower gestational weight gain (11.8 ± 5 vs 13 ± 5 kg, p = 0.05) and a higher risk of low neonate birth weight (OR 2.99, CI 95% 1.25-7.15). We conclude that the size of the FSN did not influence the maternal and neonatal outcomes...” The conclusion seems paradoxical, which seemed really confusing to me, please revise it and provide more accurate conclusions.

2.    Introduction: The logic is not clear in the introduction section, and a better organization and stronger introduction are recommended. Most of the descriptions in this section seem like unnecessary and it doesn't give sufficient information about the theme. Please focused on the significance of the size of family support network of pregnant adolescents and its probably adverse impacts on the birth outcome. Moreover, please provide more background and presented relevant results in previous epidemiological studies in this section.

3.    Materials and Methods:1) It is unnecessary to mention such lengthy context about the participants. Please re-organize those related content.

2) “The size of the FSN was divided into quartiles…” the detailed number should be present in this text as its key role in the manuscripts.

4. Results: the most of results in the manuscripts were presented by pictures, it has better also to provide the related p value in the context.

Minor comments:

1. Title: Since this paper was focused on exploring “the size of family support network”, it would be more appropriate to add the words “the size” in the title.

Author Response

The responses are included in the word document; please refer to this file. 

Reviewer 2 Report

Reviewer’s comments:

“Family Support Network of Pregnant Adolescents and Its Association with Gestational Weight Gain and Birth Weight of Newborns”

Major comment:

This study offers an interesting perspective about the relationship between family network support and birth outcomes of young adolescents. However, when reviewing the manuscript, I detected several major and minor flaws that the authors should address in their revision. My main critique concerns the language and structure of the manuscript, and the sparse use of relevant references. Many statements in the manuscript are speculative and lack (empirical) evidence. I strongly recommend the use of more references to support presented arguments. The authors might want to consider theories on social capital and theories about supportive social networks to strengthen their study.

Specific comments are listed below:

Abstract:

It should be stated in the abstract that the study was conducted in Mexico.

Line 18: Please specify “physiological/emotional changes” of pregnant women.

Line 20: It should read “The aim of the study was …”

Line 22: Revise the language: replace “including” by “referring to” or similar.

Lines 25/26: The authors should properly distinguish between findings from the descriptive analysis and findings from logistic regression. For example, the results regarding mean age were apparently not derived from logistic regression but from the descriptive statistics.

Line 32: The “quality” of the family was not analyzed. It therefore unclear how the authors come to their conclusion. I suggest to formulate a more careful conclusion (e.g. “the quality rather quantity/ network size might be more important..”

Introduction:

The introduction develops a long discussion about family network support (i.e. qualitative aspects). It seems, however, that only family size was considered in the analysis. This mismatch should be avoided or addressed early in the manuscript.

Many speculate and vague argument are presented in the introduction. Please revise and add references:

Line 37: Be more specific about “goals achievement”, some examples could be provided.

Line 42: Rather speculative statement. A reference is required.

Line 60: “paternity” – I suppose the authors mean “maternity”?

Line 65: What does “new adolescent” mean here? A reference should be added to the statement (“it has been reported…”)

Line 42: Again, a rather speculative statement. Parents’ concerns might also depend on their socioeconomic circumstances. This could be mentioned and supported with references.

Lines 84 – 90: social capital literature (alternatively studies about ‘support from social networks’) might be more appropriate here.

Material & Methods

A proper description of the family network size should appear in the method section.

Was there any information about qualitative family network aspects available in the data material? Measuring family network support based on the size of the family is rather weak instrument.

Did the authors have information about the number of children in the family (networks)?

Please provide more information about the coding of (control) variables.  

The authors should more strictly distinguish which of the findings were derived from the descriptive statistics and logistic regression. So far, the reading of the results is somewhat confusing.

Line 106: Unclear: Are the data really cross-sectional, or were repeated cross-sections used?

Lines 109 – 112: The statement about institutions is not needed and rather distracting.

Line 114: The WHO definition is not necessary.

Line 120-122: Consider a more concise formulation (for example: “…homogeneous population with hardly any immigration” ). The same applies to line 124. 

Line 147: “relationship with the world” – too trivial and general, be more concise.

Line 173: Replace “kg/M2” by “BMI<…”. According to my understanding, BMI is an index, and therefore not measured in units. Instead, a definition of “BMI” could be presented.  

Line 195: Please provide a definition of the poverty line.

Line 218: It would be better to write “5 % level”

Line 258: The odds rations here indicate a three-fold higher risk.

Discussion

Again, many statement and conclusion are rather lengthy and could be cut down. Be more concise and avoid long sentences.

Lines 308 – 311 for example, please revise the (too) long sentence.

Lines 300 – 304: split the lengthy sentence.

Line 289: What are “healthy outcomes”?

Line 293: In what way are instruments different?

Line 324: What are these other disadvantages?

As mentioned before, it seems there is some “overinterpretation” of the findings – the quality of family network support was not formally. Please choose a more careful formulation in the conclusions.

Author Response

The responses are included in the Word document; please refer to this file. 

Round  2

Reviewer 1 Report

The authors have properly replied to all comments.

Author Response

Thank you very much for your comments

Reviewer 2 Report

Reviewer’s comments

The authors’ revisions substantially improved the manuscript. I found still some errors that should be corrected. Also, I recommend a spell and grammar check of the entire manuscript by a native speaker. Some examples of errors can be found below.

Line 21/22: The statement is inappropriate and needs to be justified, e.g. by writing “previous research has shown that the type and number of members in FSN have positive effects…”. Make sure that there is literature/ research supporting the argument.

Line 180: Please correct, it should read: “We observe that it was common…”

Line 245: strange wording. Please replace “determined” by “demonstrated” or similar.

Line 254: Typing error: Elsenbruch

Line 257: “… there was a lower gestational weight gain…”

Author Response

Thanks very much for your comments, to answers to each of the observations will be found in the attached file
